# Most Plant Nutrient Elements Are Retained by Biochar in Soil

**Tanawan Limwikran [1], Irb Kheoruenromne [1],\*, Anchalee Suddhiprakarn [1], Nattaporn Prakongkep [2] and Robert J. Gilkes [3]**

[1] Department of Soil Science, Faculty of Agriculture, Kasetsart University, Bangkok 10900, Thailand; tanawan.lim@gmail.com (T.L.); agrals@ku.ac.th (A.S.)

[2] Office of Science for Land Development, Land Development Department, Bangkok 10900, Thailand; asoil@hotmail.com

[3] School of Agriculture and Environment, Faculty of Science, University of Western Australia, Crawley 6009, Australia; bob.gilkes@uwa.edu.au

\* Correspondence: irbs@ku.ac.th

**Abstract:** Biochar may contain substantial amounts of plant nutrient elements, and at typical rates of application, may supply luxury levels of K, Ca, P, and other plant nutrients. However, little is known of the agronomic effectiveness of these nutrients because they exist in diverse compounds and are located in the microporous matrix of biochar particles. We have identified the compounds and location of nutrient elements in three biochars and observed their release from biochar particles in soil. Much K was quickly released from biochar but little or no Ca, Mg, S, and P were released over eight months, which represents a very different behavior from chemical fertilizers that are mostly water soluble. There is clearly a need to determine the availability to plant nutrients in biochar. Appropriate laboratory methods should be developed for measuring the availability of plant nutrients as standard methods of fertilizer analysis are ineffective.

**Keywords:** biochar; plant nutrients; dissolution; microporosity

## 1. Introduction

Biochar may contain substantial amounts of plant nutrient elements, and at normal rates of application of biochar to crops, it may act as a multi-element fertilizer [1,2]. The solubility in extractants of the nutrient compounds (minerals) within biochar and dissolution of these minerals in soils is commonly low due to the intrinsic properties of the minerals and their protected positions within the microporous fabric of biochar [3,4]. The relative effectiveness (RE) and residual value (RV) of most nutrients supplied in biochar will be less than for the corresponding water soluble chemical fertilizers (e.g., muriate of potash (KCl), monocalcium phosphate) [5]. Farmers need to be informed of RE and RV values for particular combinations of biochars, soils, agronomic practices, and crop species to allow them to identify if biochar can be used to replace or supplement conventional fertilizers. Such information can only be obtained from long term and expensive field experiments that is rarely available. An indication of RE and RV values of nutrients in biochar can be obtained from laboratory experiments that measure the release of nutrients from biochar to soil [4]. These experiments show that both the extent and rate of dissolution of K, Ca, and P from biochar in soils may be limited and cannot be predicted on the basis of soil properties. Clearly, knowledge of the identity of minerals containing these and other plant nutrient elements together with their behavior in soil is required. To obtain this information, we incubated three biochars (mangosteen shell, pineapple peel, and eucalyptus wood) in nine tropical soils for periods up to 8 months. The recovered biochar particles were analyzed for

total and water-soluble K, Ca, Mg, Mn, P, and S. Particles were fractured to allow analysis of particle interiors by SEM/energy dispersive X-ray spectrometry (EDS), which identified residual undissolved minerals and minerals that had migrated from soil into biochar particles. This research examines the hypothesis that release of plant nutrient elements from within biochar particles in soil is affected by the diverse compounds containing elements and their location in biochar particles.

## 2. Materials and Methods

### 2.1. Soil Samples

Soil samples were taken from the surface (0–20 cm depth) of nine soil profiles from Nakhon Ratchasima province in Thailand. These soils are classified as Oxisols and Ultisols [6]. Soil series [4], soil classification [6], parent material, and location are listed in Table 1. Soil samples were air-dried, gently crushed, and passed through a 2 mm sieve before analysis. Physical and chemical properties of the soils are shown in Table 2.

**Table 1.** Soil series, location, classification, and parent material of soils used in the study.

| Soil Series [4] | Coordinates | Classification [6] | Parent Material |
|---|---|---|---|
| Oxisols | | | |
| Pak Chong1 (Pc1) | 47P 0,759,730 E 1,624,306 N | Rhodic Kandiustox | Residuum and colluvium from limestone |
| Pak Chong2 (Pc2 | 47P 0,758,389 E 1,616,034 N | Rhodic Kandiustox | Residuum and colluvium from limestone |
| Chok Chai (Ci) | 48P 0,210,727 E 1,633,381 N | Rhodic Kandiustox | Residuum and colluvium from basalt |
| Ultisols | | | |
| Roi Et1 (Re1) | 47P 0,795,612 E 1,680,167 N | Aeric Kandiaquults | Wash deposit from sandstone |
| Roi Et2 (Re2) | 47P 0,788,656 E 1,652,504 N | Aeric Kandiaquults | Wash deposit from sandstone |
| Khorat (Kt) | 47P 0,792,694 E 1,678,613 N | Typic Kandiustults | Wash deposit from sandstone |
| Yasothon (Yt) | 47P 0,793,502 E 1,675,845 N | Typic Paleustults | Wash deposit from sandstone |
| Sung Noen1 (Sn1) | 47P 0,791,130 E 1,666,752 N | Typic Paleustults | Wash deposit over siltstone/shale |
| Sung Noen2 (Sn2) | 47P 0,790,298 E 1,664,657 N | Typic Paleustults | Wash deposit over siltstone/shale |

**Table 2.** Physicochemical properties of the soils used in the study.

| Soil Series [4] | pH | EC | Clay | OC | CEC | K | Na | Ca | Mg | Avail. P |
|---|---|---|---|---|---|---|---|---|---|---|
| | | µS cm$^{-1}$ | (%) | | cmol kg$^{-1}$ | | (cmol kg$^{-1}$) | | | mg kg$^{-1}$ |
| Oxisols | | | | | | | | | | |
| Pc1 | 6.4 | 53 | 32 | 0.66 | 12 | 2.1 | 0.0 | 5.2 | 2.1 | 76 |
| Pc2 | 4.8 | 51 | 36 | 0.54 | 12 | 0.3 | 0.0 | 4.1 | 1.4 | 65 |
| Ci | 5.0 | 61 | 32 | 0.47 | 10 | 0.3 | 0.1 | 2.3 | 0.5 | 47 |
| Ultisols | | | | | | | | | | |
| Re1 | 6.3 | 199 | 20 | 0.42 | 12 | 0.5 | 0.1 | 5.7 | 2.1 | 113 |
| Re2 | 5.4 | 73 | 24 | 0.41 | 11 | 0.5 | 0.1 | 5.3 | 0.7 | 18 |
| Kt | 5.2 | 81 | 12 | 0.27 | 6 | 0.2 | 0.1 | 2.3 | 1.1 | 26 |
| Yt | 5.1 | 35 | 8 | 0.13 | 3 | 0.2 | 0.0 | 0.9 | 0.2 | 15 |
| Sn1 | 4.7 | 86 | 28 | 0.40 | 12 | 0.9 | 0.1 | 4.3 | 1.6 | 41 |
| Sn2 | 5.8 | 94 | 20 | 0.40 | 10 | 0.5 | 0.2 | 3.3 | 1.3 | 34 |

pH and EC by 1:5 H$_2$O extraction. EC, electrical conductivity. OC, organic carbon. CEC, cation exchange capacity, Avail. P, available phosphorus.

## 2.2. Preparation of Biochars

Mangosteen shell, pineapple peel, and eucalyptus wood wastes were chosen for this research as they exist in large volumes in many tropical countries [2]. They were dried to 20–30% moisture before the carbonization procedure. Biochar was produced in a 200 L cylindrical container filled with the dried wastes [2]. The carbonization temperature was measured using a thermocouple and maintained at 400 °C with a 5 h residence time. This temperature corresponds to conditions employed in traditional biochar ovens operated by farmers in Thailand [2]. The biochar samples were allowed to cool to room temperature, then were crushed and sieved to a size range of 2 to 4 mm for the incubation experiment. Properties of the biochars are shown in Tables 3 and 4.

**Table 3.** The pH, EC, and total element concentrations of biochars [a].

| Raw Materials | pH | EC | Ca | K | Mg | Mn | Na | P | S |
| | | $\mu S\ cm^{-1}$ | | | | $(mg\ kg^{-1})$ | | | |
| --- | --- | --- | --- | --- | --- | --- | --- | --- | --- |
| Mangosteen shell | 8.2 | 6070 | 4796 | 28,758 | 1084 | 97 | 388 | 10,346 | 1036 |
| Pineapple peel | 7.9 | 13,273 | 6466 | 32,528 | 2231 | 235 | 333 | 4211 | 1555 |
| Eucalyptus wood | 10.2 | 2677 | 14,362 | 5552 | 944 | 1273 | 1001 | 3463 | 261 |

[a] Biochar samples finely ground and digested in 5:2 mixture of $HNO_3/HClO_4$, measured by inductively couple plasma optical emission spectrometer (ICP-OES), pH and EC by 1:5 $H_2O$ extraction.

**Table 4.** The proportions of water soluble elements in biochars [a] and the molar ratio of dissolved K/Cl.

| Raw Materials | Ca | K | Mg | Mn | Na | P | S | K/Cl |
| --- | --- | --- | --- | --- | --- | --- | --- | --- |
| Mangosteen shell | 0.05 | 0.26 | 0.01 | 0.00 | 0.23 | 0.03 | 0.17 | 0.69 |
| Pineapple peel | 0.07 | 0.54 | 0.02 | 0.00 | 0.24 | 0.17 | 0.63 | 0.87 |
| Eucalyptus wood | 0.04 | 0.42 | 0.20 | 0.00 | 0.30 | 0.02 | 0.30 | 1.57 |

[a] Ground samples were extracted by Milli-Q (MQ) water for 16 h with shaking, and analyzed by ICP-OES.

## 2.3. Incubation Experiment

The incubation experiment was carried out in the School of Agriculture and Environment, University of Western Australia laboratory using rectangular plastic 310 $cm^3$ containers. The method had been developed and evaluated in an earlier experiment [4] and duplicated treatments in the second experiment produced near identical results. A 100 g soil sample (<2 mm particle size) was placed in the container with 1 g of 2–4 mm particles of biochar placed at 2 cm depth, sandwiched between two sheets of 1 mm nylon mesh and two equal layers of soil. This arrangement provided intimate contact between the biochar and soil, enabling the soil solution, soil particles, and fungal hyphae to enter the biochar particles. The treatments were replicated four times (for 1, 2, 4, and 8 month incubations) and incubated in the dark at 25 °C at 90% field capacity, which was maintained by adding deionized water gently to the soil surface every 3 days. After incubation, biochar particles were separated from the soil and dried at room temperature.

## 2.4. Chemical Analysis

The original and recovered biochars were ground and analyzed for pH, electrical conductivity (EC), water soluble elements, and total elements. Biochar pH and EC were measured using a 1:66 solid/Milli-Q (MQ) water extraction with shaking end-over-end for 16 h. The total elements in the biochars were determined by digesting 0.3 g ground biochar in 5 mL of a mixture of concentrated nitric acid ($HNO_3$) and perchloric acid ($HClO_4$) [7]. The filtered digests were analyzed by atomic absorption spectroscopy (AAS). The water soluble elements in biochar were determined by extracting the ground biochar with MQ water (0.3 g biochar per 10 mL MQ water) and shaking end-over-end for 12 h. The filtered extracts were analyzed by inductively couple plasma optical emission spectrometer

(ICP-OES). Water soluble P and total P were determined colorimetrically using a spectrophotometer. The soil samples were analyzed for pH and EC in a 1:5 soil/water extract. Exchangeable K, Na, Ca, and Mg were measured by the ammonium saturation method at pH 7.0 [8]. Available phosphorus was determined by the Colwell method, using an extracting solution of 0.5 M $NaHCO_3$ adjusted to pH 8.5 with NaOH, a soil solution ratio of 1 g soil/25 mL $NaHCO_3$ solution, and an extraction time of 16 h at 25 °C [9]. Phosphorus was determined colorimetrically.

## 2.5. XRD

Compounds (minerals) in the biochars were identified by X-ray diffraction analysis using an XPert$^3$ powder diffractometer with a Ni filter (CuKα, 45 kV, 40 mA) and Pixel detector. Ground biochars (size < 0.05 mm) were scanned from 5 to 70° 2θ, using a step size of 0.015° 2θ and a scan speed of 0.02° 2θ s$^{-1}$.

## 2.6. SEM, EDS, and X-ray Mapping

To investigate the interior of original and recovered biochar particles, several particles for each biochar were fractured and placed fracture face uppermost on a conductive carbon tape stuck to an aluminum stub and were coated with carbon prior to examination. Biochar morphology was observed by scanning electron microscopy (SEM) of the biochar fracture surfaces. A TESCAN VEGA3 SEM was operated at an acceleration voltage of 15 kV, and EDS (energy dispersive spectrometry of X-rays) provided rapid, semi-quantitative analyses of the elemental composition of particles on the fracture surface. Several regions were examined for each biochar particle.

Element distributions were mapped by two-dimensional scanning of areas of interest using an X-ray dot mapping procedure, which involves taking the output of a single channel analyzer (SCA) tuned to a single element and using it to modulate the brightness of the screen during normal electron raster scanning. Each X-ray photon detected appears as a dot on the screen, with regions of high concentration characterized by a high dot density [10]. The net intensity measured for each element was recorded for each scanned area and used to provide a semi-quantitative analysis of the area. Spectra were obtained from several fracture surfaces for individual biochar particles recovered from all nine soils. Integrated analyses of large areas (ca. 0.16 mm$^2$) were obtained and these analyses have been expressed as percentages of the total intensity due to the major cations and anions detected by EDS. Intensities due to carbon and oxygen have been omitted from this analysis.

## 3. Results

### 3.1. Properties of Biochars

The chemical properties of the two fruit waste biochars (mangosteen shell and pineapple peel) were similar, as shown in Table 3. The pH was slightly alkaline and EC values were high, indicating that these biochars contained much soluble salts. Total K concentrations were high and total Ca, Mg, P, and S concentrations were moderate to high. Eucalyptus wood biochar differed considerably from fruit waste biochars as it was highly alkaline, had low EC, much less K, and more Ca.

The proportions of elements soluble in water, as shown in Table 4, were quite diverse for these biochars. Little (<21%) of the Ca, Mg, and P was soluble, whereas larger to major proportions (26–63%) of the K and (0.17–63%) of the S were soluble. The K/Cl ratio (0.69, 0.87) in the water extract for fruit waste biochars was consistent with much of the soluble K being in water-soluble KCl, whereas the higher ratio (1.57) for eucalyptus wood biochar indicates that some of the soluble K was not in KCl.

X-ray diffraction patterns, as shown in Figure 1, of original biochars show that mangosteen shell and pineapple peel biochars contain sylvite (KCl), which is consistent with their high K concentrations (2.9%, 3.3%, respectively). Sylvite was not detected in the eucalyptus wood biochar. Calcite ($CaCO_3$) was abundant in eucalyptus wood biochar [11], which is consistent with its high Ca (1.4%) content. No XRD peaks for other K or Ca minerals or for Mg, Mn, P, or S minerals were detected, which is partly a

consequence of the low sensitivity of the XRD technique and the high background due to amorphous carbon [11]. It must be noted that water-soluble elements were determined in finely ground biochar and would not indicate the elements that would dissolve from biochar particles.

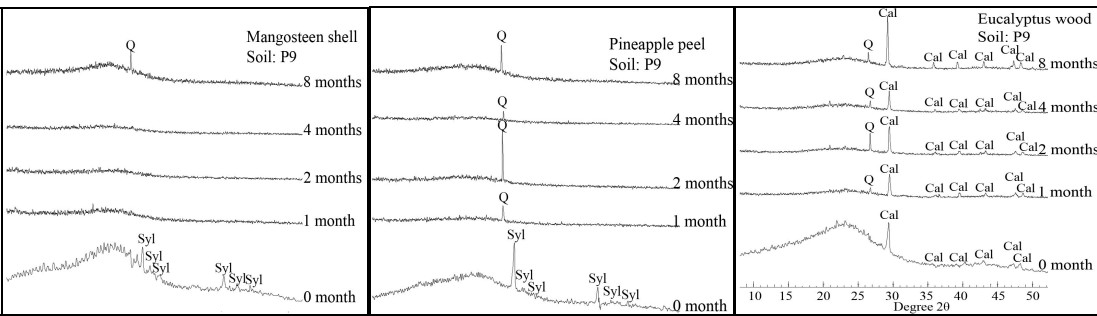

**Figure 1.** X-ray diffraction patterns of mangosteen shell, pineapple peel, and eucalyptus wood biochars for original biochar and biochar recovered from soil (P9) after 1, 2, 4, and 8 months. The vertical axis (intensity) scale differs between patterns.

XRD patterns of mangosteen shell and pineapple peel biochars particles recovered after various times in the soil show that all sylvite had left the biochars within one month. The calcite in eucalyptus wood biochar remained after eight months. All biochars had acquired quartz from the soil during incubation.

*3.2. Changes in the Chemical Composition of Biochar during Incubation*

Changes in the amounts of soluble and insoluble elements in biochars during the eight months of incubation are shown in Figures 2–4. The points in the graphs represent average data for the nine soils with standard deviation values shown as error bars. The error bars are small, indicating that there were no major differences in amounts of elements lost for the different soils.

For all three biochars, most of the soluble and insoluble K had been lost from the particles within the first month of incubation in soil, with continuing losses up to eight months. Readers should note that the term insoluble K indicates that K was not extracted from ground biochar during an extraction in water possibly because it was protected from dissolution within the microporous fabric of biochar. This insoluble K may be present in water-soluble compounds, but these are located in protective pores within the microporous fabric of biochar. The same nomenclature is used for other elements [12].

Amounts of insoluble Ca increased moderately or remained constant during incubation, whereas amounts of water soluble Ca increased greatly, although remaining much smaller than the amounts of insoluble Ca. Thus, the biochars had absorbed Ca from the soils. The soils contained considerable exchangeable Ca, as shown in Table 2, some of which may have been replaced by K released from biochar. Insoluble Mg remained almost constant for all biochars over the eight month incubation period, as shown in Figure 3. Water soluble Mg increased greatly for mangosteen shell and pineapple peel biochars, but remained almost constant for eucalyptus wood biochar, as shown in Figure 3. Water-soluble Mg was much less than insoluble Mg for all biochars, so some of this additional water soluble Mg could have been derived from the initially insoluble Mg fraction rather than by sorption from soil. Insoluble Mn decreased greatly within one month for mangosteen and eucalyptus biochars, while remaining almost constant for pineapple peel biochar. Water soluble Mn increased over time for all three biochars but was a much smaller amount of Mn than insoluble Mn. Much of the initially insoluble and soluble P remained in all three biochars after eight months of incubation, as shown in Figure 4. Much of the insoluble S remained in mangosteen shell and pineapple peel biochars, whereas insoluble S increased greatly for eucalyptus wood biochar. Water soluble S remained almost constant for mangosteen shell and eucalyptus wood biochars and decreased greatly for pineapple peel biochar.

The results shown in Figures 2–4 resemble those reported from a similar experiment by Limwikran et al. [4], for which incubation of the same biochars was limited to two months. In the present work, the much longer (8 months) incubation time has resulted in greater losses of soluble K and insoluble K from all three biochars. The gains in water soluble and insoluble Ca by all three biochars continued from two to eight months. The losses of insoluble P and the small change in soluble P for mangosteen shell biochar persisted for 8 months. For pineapple peel biochar, the relatively minor loss of insoluble P and moderate loss of soluble P continued from two to eight months. For eucalyptus wood biochar, the absence of systematic changes in soluble and insoluble P concentrations persisted over 8 months.

The divergent behavior of different elements and of different biochars is probably associated with the various minerals containing these elements that are present in biochar, as is investigated below. It is evident that the limited release of some plant nutrient elements from biochar to soil is quite different from the rapid loss of most K and P from chemical fertilizers [13–15].

The slower release of nutrients from biochar might be beneficial in sandy soils that experience rapid leaching of nutrients from the root zone, but in general, the limited release over eight months of several nutrient elements is likely to greatly reduce the effectiveness of biochar as a fertilizer.

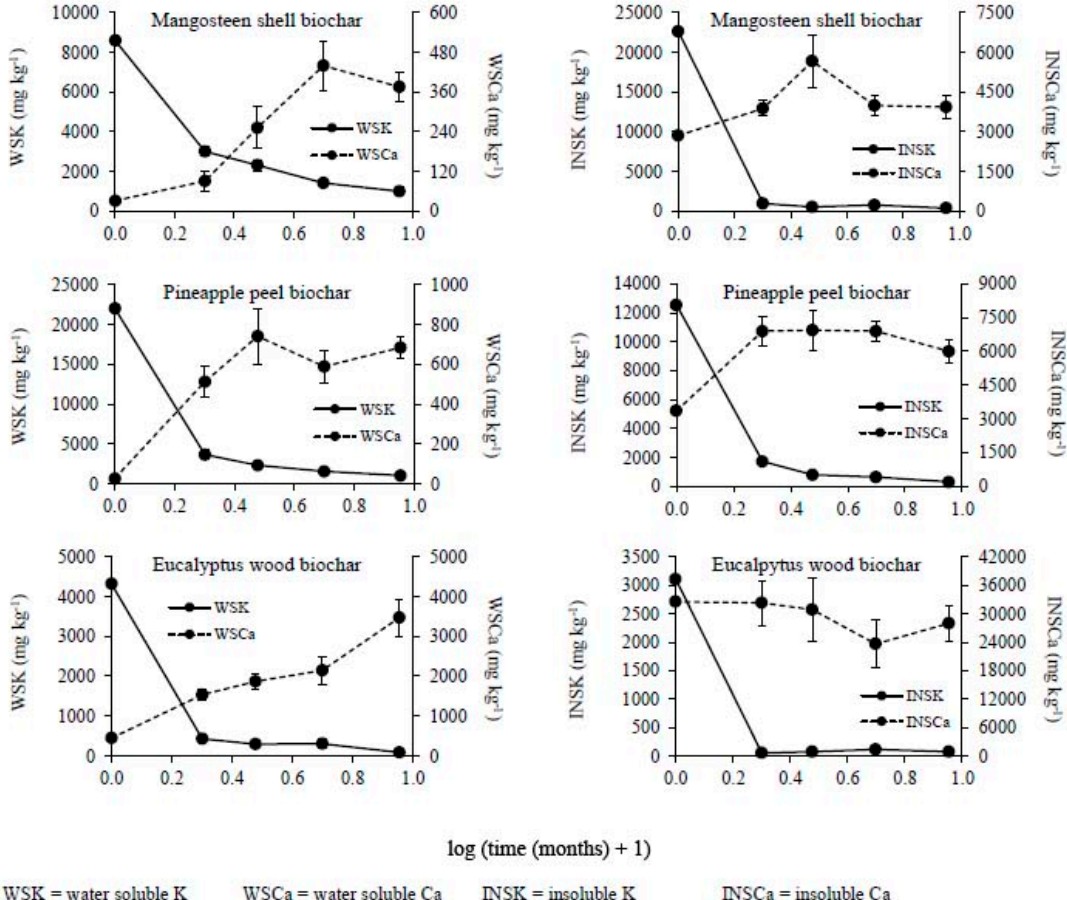

**Figure 2.** Plots of mean value and standard deviation for water soluble and insoluble K and Ca versus time in soil for three biochars recovered from soil at zero, 1, 2, 4, and 8 months. Mean values and SD are for nine soils.

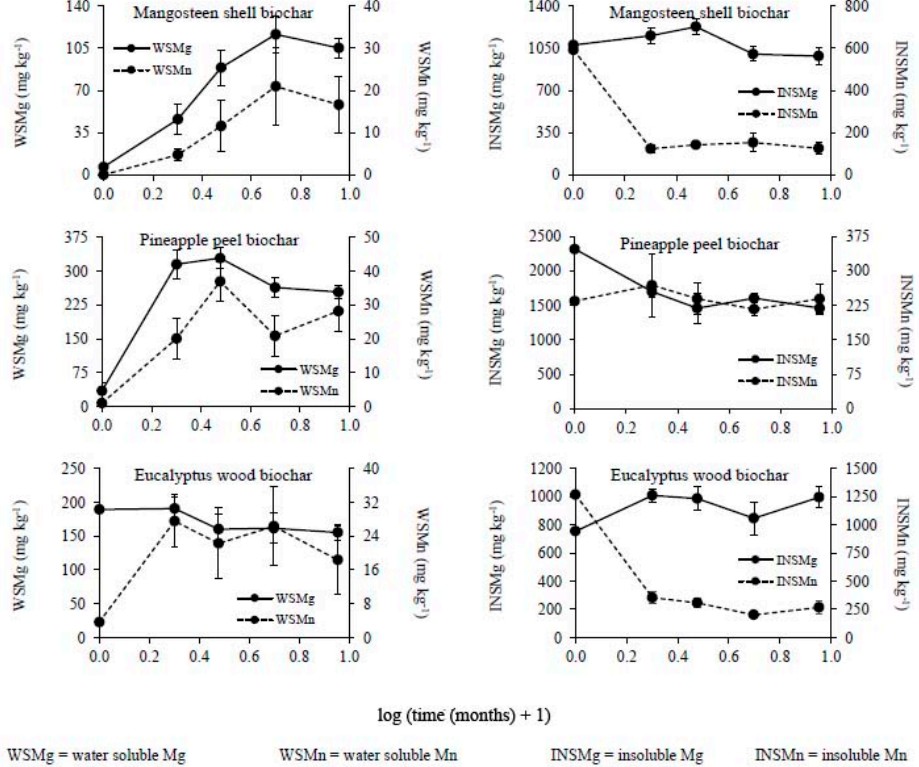

**Figure 3.** Plots of mean values and standard deviation for water soluble and insoluble Mg and Mn versus time in soil for three biochars recovered from soil at zero, 1, 2, 4, and 8 months. Mean values and SD are for nine soils.

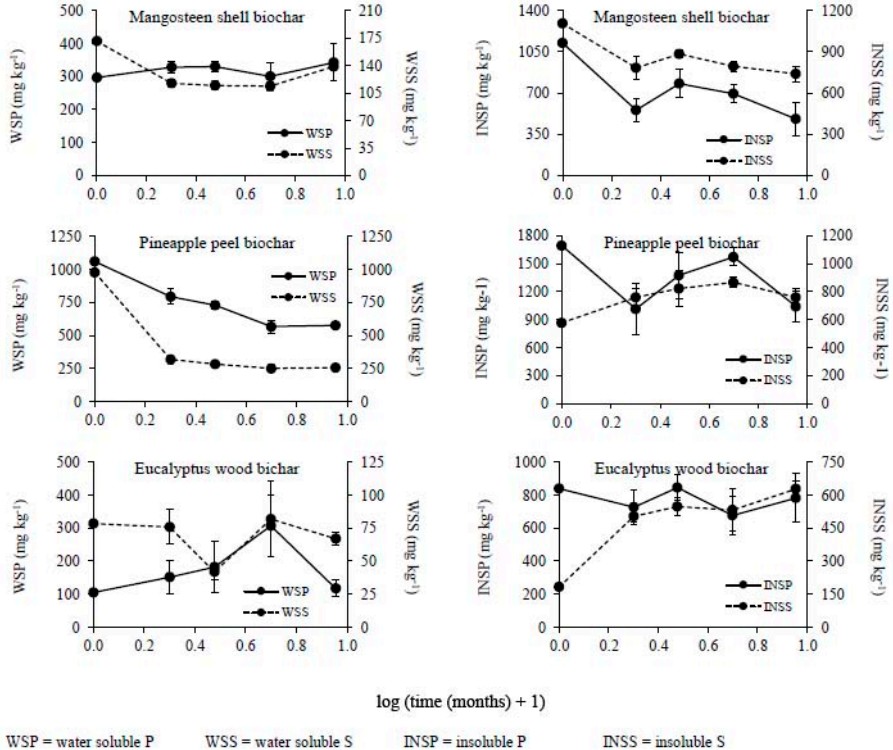

**Figure 4.** Plots of mean value and standard deviation for water soluble and insoluble P and S versus time in soil for three biochars recovered from soil at zero, 1, 2, 4, and 8 months. Mean values and SD are for nine soils.

### 3.3. SEM/EDS Analyses of Biochar Particles Recovered from Soil

The diverse dissolution behaviors of plant nutrient elements within biochars incubated in soil are at least partly due to the various mineral forms of these elements and their locations within the microfabric of biochar particles. These two aspects were investigated by scanning electron microscopy (SEM) and energy dispersive X-ray spectrometry (EDS) analysis of the interior of recovered biochar particles after zero, one, two, four, and eight months of incubation in the soil. Average data for nine soils and three biochars after zero, one, two, four and eight months of incubation are shown in Figure 5. Spectra were quite variable both for different fracture surfaces of individual biochar particles and biochars from different soils. Consequently, the error bars in Figure 5 are often quite large, but the major trends are still clear. For all three biochars, the relative abundance of K had decreased greatly during one month of incubation, with further losses up to eight months of incubation. This trend mirrors the changes in insoluble and soluble K discussed earlier, as shown in Figure 2. The rapid decrease in the abundance of K is matched by the decrease in Cl for pineapple peel and mangosteen shell biochars and is associated with the dissolution of sylvite (KCl), the mineral identified by XRD analysis, as shown in Figure 1. Eucalyptus wood biochar also lost much K, but there was no associated loss of Cl, which is consistent with XRD not detecting sylvite in this biochar, and K was mostly present in other minerals [2]. Much of the K remaining in all three biochars after one month was likely to be in one or more of the moderately to poorly soluble K-compounds that are common constituents of burnt plant materials (e.g., fairchildite ($K_2Ca(CO_3)_2$), archerite (($K,NH_4)H_2PO_4$), pyrocoproite (($Mg(K,Na))_2P_2O_7$) [16]). These minerals generate weak XRD patterns so would not be evident in Figure 1, but highly sensitive synchrotron XRD has been used by other researchers to identify pyrocoproite and archerite in biochars produced from tropical plant wastes [2].

For mangosteen skin and pineapple peel biochars, the relative amounts of Ca and Mg exposed on fracture surfaces increased after two months of incubation, which is consistent with the values of insoluble and water soluble Ca and Mg shown in Figures 2 and 3. Eucalyptus wood biochar contained much Ca, and its relative abundance on fracture surfaces decreased with incubation period due at least partly to increasing amounts of clay (Al, Si, Fe) and quartz (Si) diluting the observed concentration of Ca. The relative amount of Mg remained almost constant as did insoluble and soluble Mg, as shown in Figure 3.

For pineapple peel biochar, the relative amounts of P and S determined by EDS analysis of fracture surfaces remained approximately constant as did insoluble P and S, as shown in Figure 4; although, water soluble P and S, which represent about half of the P and S in pineapple peel biochar, decreased in abundance. For mangosteen shell biochar, the relative amounts of P and S determined by EDS on fracture surfaces had increased after eight months; whereas insoluble P and S had decreased to a moderate extent and water soluble P and S (minor proportions) remained approximately constant, as shown in Figure 4.

For eucalyptus wood biochar, the relative amount of S on fracture surfaces decreased; whereas the insoluble S increased over time, as shown in Figure 4. It is likely that the high initial S value (5%) calculated from EDS spectra for original eucalyptus wood biochar is too high and the variation in this value for different surfaces is very large, indicating a large range of S concentration existed (±3%). There was no systematic trend over time for the amount of P in fracture surfaces of eucalyptus wood biochar, which is consistent with the amounts of insoluble and soluble P present in this biochar, as shown in Figure 4.

We conclude, from these integrated EDS analyses of the many compounds on fracture surfaces of fruit waste biochars recovered from soils, that rapid dissolution of sylvite (KCl) had greatly reduced the amounts of K and Cl in mangosteen shell and pineapple peel biochar particles. Much K was also lost from eucalyptus wood biochar, but there was no associated loss of Cl, so K compounds other than sylvite had dissolved. At the same time, the amounts of Ca and Mg increased in pineapple peel and mangosteen shell biochars due to migration of these elements from soil into particles. The amount of Ca in eucalyptus wood biochar was initially high and decreased slightly with incubation. Phosphorus

and S were largely retained within fruit waste biochar particles, but eucalyptus wood biochar lost much S. Silicon, Al, and Fe concentrations in particle fracture surfaces increased greatly during incubation, as fine soil particles migrated through pores and cracks into the interior of biochar particles. Further explanations for these diverse behaviors were sought through SEM analysis of individual mineral particles and mineral mixtures within biochar particles.

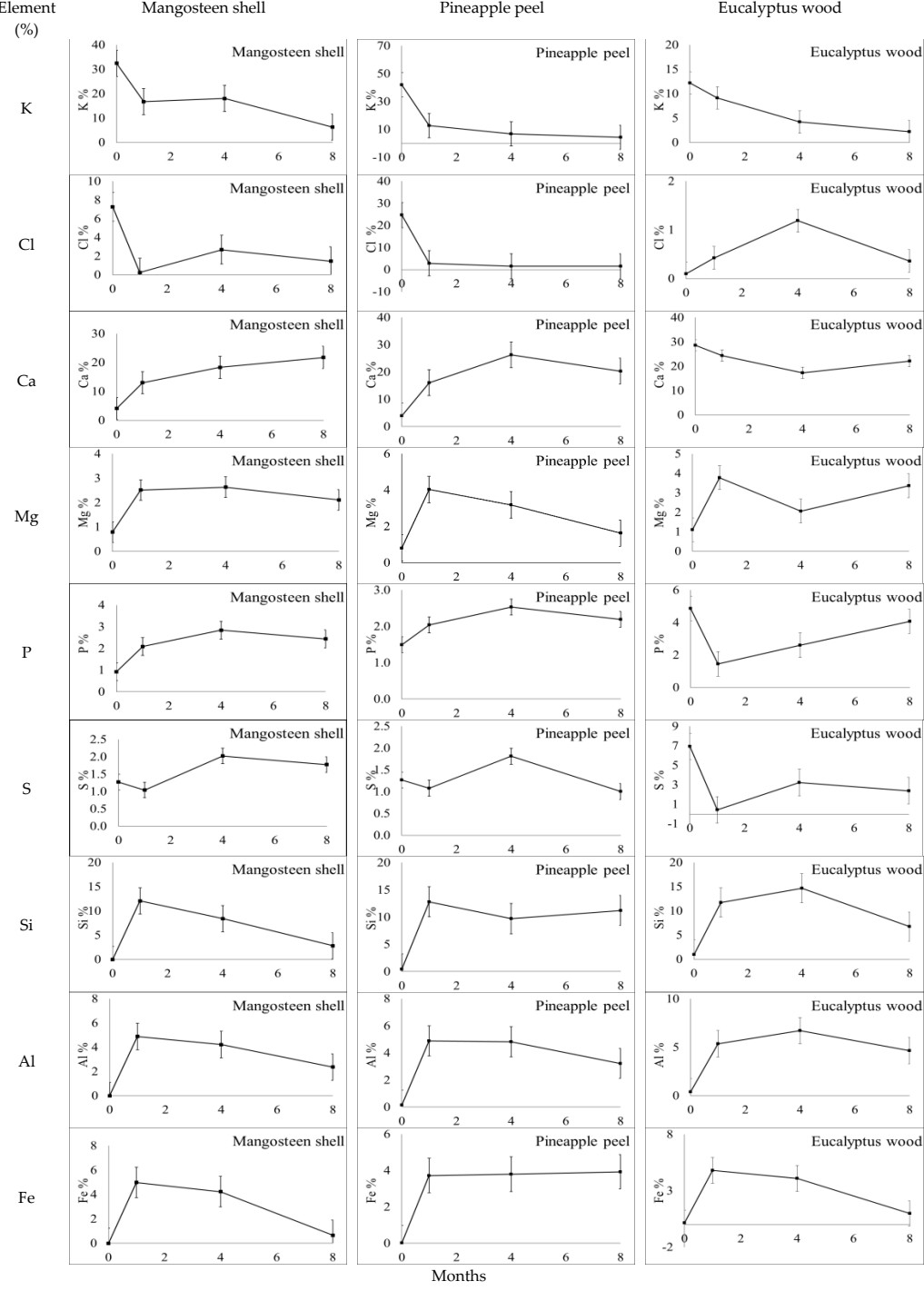

**Figure 5.** Semiquantitative elemental analyses of fracture surfaces of three biochars recovered after zero, 1, 2, 4, and 8 months of incubation in the soil. Values are average and SD for nine soils. The lower limit of SD values extend to negative values for K and Cl in pineapple peel biochar, which is a consequence of the SD values being large compared to the average values.

### 3.4. Point and Particle Analyses of Minerals in Biochar

SEM images of fracture surfaces are dominated by the complex fabric of biochar carbon, which faithfully preserves the diverse tissues in the parent plant materials, as shown in Figures 6 and 7. Tissues may have contained solutions that precipitated into various compounds when the plant material was air dried, and these compounds may have persisted or altered during heating to create biochar. For example, some of the K in solution in stem exudate and xylem sap is present as dissolved KCl [17], although for tree tobacco (*Nicotiana glauca*) concentrations of 3673 and 204 µg/mL K and 486 and 64 µg/mL Cl, respectively, indicate that only a minor proportion of the K would crystallize as KCl (sylvite). Micron-sized crystals of sylvite would have precipitated on the surfaces of cells during drying of cell solutions. Figure 6a shows sylvite in nonincubated mangosteen biochar. These crystals seem to be unaffected by biochar manufacture, which converted cell walls to carbon. Calcium is incorporated in several plant constituents [18], including cell walls, and also occurs as crystalline calcium oxalate within cells [19]. Heating to produce biochar at temperatures above about 400 °C dehydrates calcium oxalate to produce porous calcium carbonate particles which often preserve the shape of the parent calcium oxalate particle, as is seen in Figure 6b [20]. During heating of plant material, the Ca in cell walls reacts with associated elements to form diverse compounds and especially calcium phosphate minerals, as shown in Figures 6d and 7a.

Fracture surfaces of biochar particles recovered from soil commonly contain aggregates of soil clay with diverse compositions representing various mixtures of kaolin, illite, and iron oxides mixed with Mg/K salts derived from biochar, as shown in Figure 6c. Silt-size particles of quartz with attached clay also occur on fracture surfaces, which represent the surfaces of cracks that provided a pathway for silt size particles to migrate into biochar from the soil, as shown in Figure 7c. Phosphorus is present in plant materials within several organic compounds, including pectates and proteins, which are oxidized by heating and lose their volatile constituents (C, N), so that calcium phosphate compounds remain as discrete particles within pores in biochar, as is shown for eucalyptus wood biochar in Figure 6d. Calcium phosphates are also finely disseminated throughout former cell walls, as shown in Figure 7a.

Calcium phosphate compounds remain as disseminated particles within former cell walls, as shown in Figure 7a, and as discrete particles in pores, as is shown for eucalyptus wood biochar in Figure 6d. The composition of the particle shown in Figure 6d for incubated eucalyptus wood biochar is $Ca_{3.96}Mn_{0.66}Mg_{0.48}(PO_4)_3OH$, which approximates the formula of hydroxyapatite ($Ca_5(PO_4)_3OH$) [21]. Many discrete P-rich particles in biochar have similar compositions to this example, but other Ca, Mn, and Mg/P ratios also occur. The disseminated calcium phosphate accumulation within the porous carbon replacing cell wall, shown in Figure 7a, is from pineapple peel biochar that was recovered from soil after 8 months. It has the approximate composition $Ca_{4.9}Mn_{0.1}Mg_{0.1}(PO_4)_3OH$ and so quite closely resembles ideal apatite. The apatite structure is able to accommodate a wide range of cations [22] and this capacity is evident in the various cation contents (Ca, Mg, Mn) of apatite particles present in biochar. The persistence of apatite in biochar in soil is to be expected, as apatite fertilizers often persist in soils for long periods as they require an acid soil environment to promote dissolution [23]. The interior of biochar particles in soil is commonly alkaline, so that apatite will not dissolve until soil acidity has consumed the alkalinity within biochar particles, which is a slow process. Calcium sulphate-rich particles occur in biochar, as for example the particle in pineapple peel biochar recovered from soil after four months shown in Figure 7b, with the approximate formula of $Ca_3SO_4$. This formula is not balanced, so the particle is probably a 1:2 mixture of $CaSO_4$ (anhydrite) and $CaCO_3$ (calcite). Indeed, many mineral particles in biochar are fine particle mixtures of several compounds, which consequently have complex X-ray spectra. Figure 7d shows the spectrum of material composed mostly of K (45 percent) and Ca (18.5 percent) with minor amounts of S, Mg, and P. The dominant compound may be fairchildite ($K_2Ca(CO_3)_2$), a common constituent of plant ash [16,24].

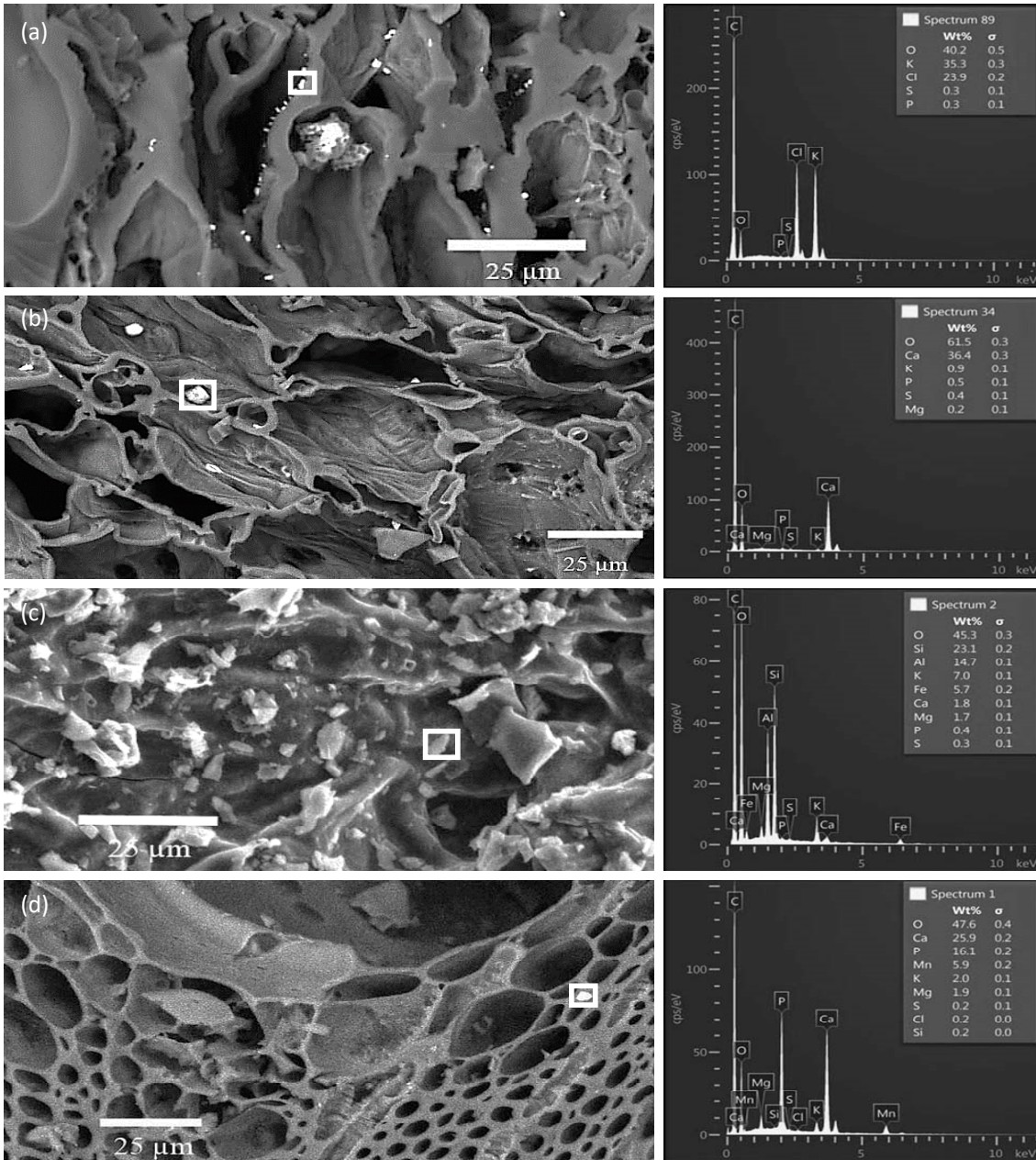

**Figure 6.** SEM images and spectra of particles on fracture surfaces of original mangosteen biochar (**a**), mangosteen biochar recovered from soil after 8 (**b**) and 1 (**c**) months. Eucalyptus biochar recovered from soil after 8 months (**d**). Energy dispersive X-ray spectrometry (EDS) spectra and analyses of the indicated mineral particles are shown.

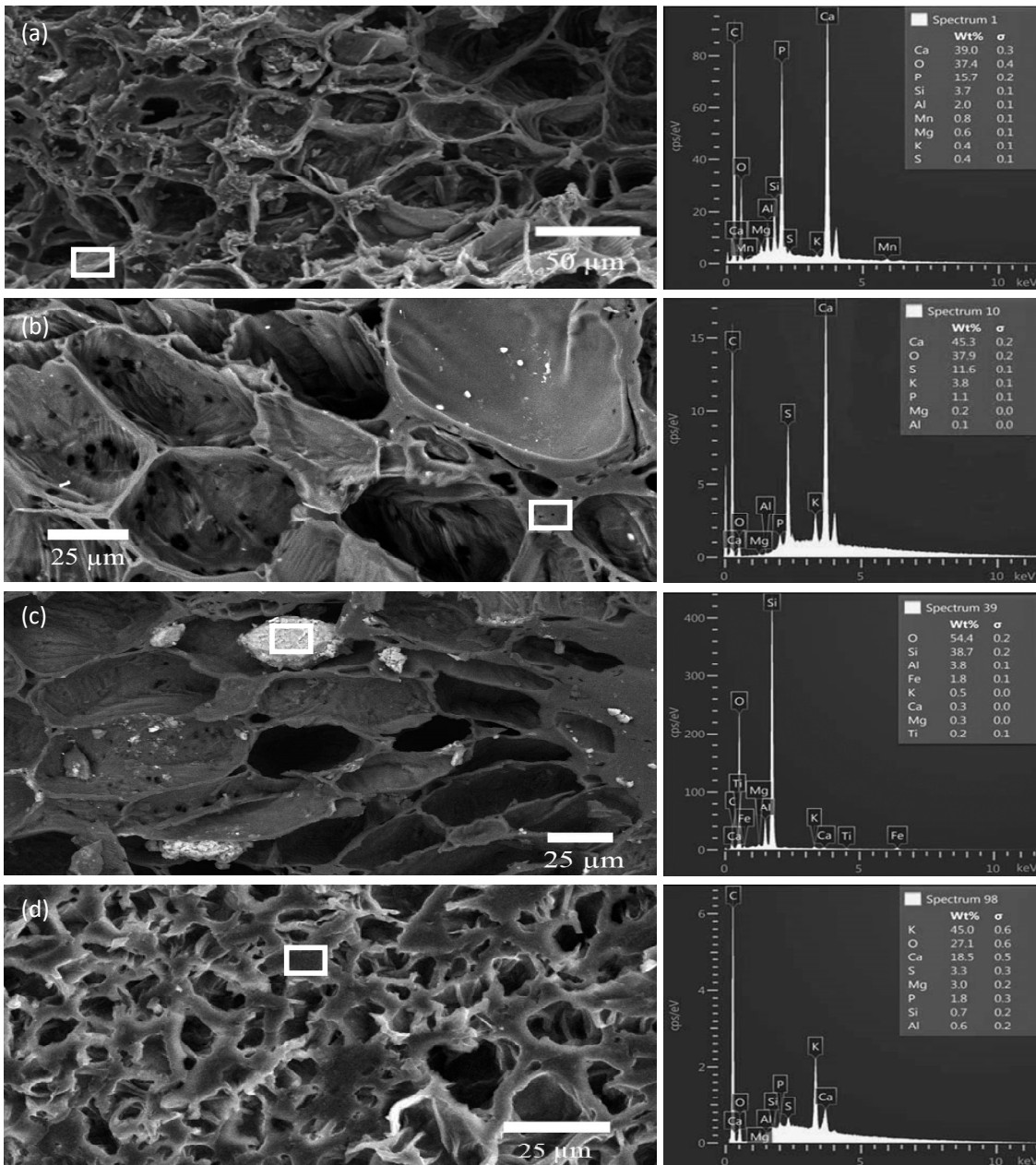

**Figure 7.** SEM images and spectra of particles and matrix fracture surfaces of pineapple peel biochar recovered from soil after 8 (**a**), 4 (**b**), 8 (**c**) and 1 (**d**) months. EDS spectra and analyses of the indicated mineral particles (**c**) or matrix (**a**,**b**,**d**) are shown.

The wide range of compositions of particles in biochar is evident when the point analyses of particles or mineralized regions of cell walls exposed in fracture surfaces are plotted in factor diagrams, as shown in Figure 8. These diagrams contain data for many particles and regions (henceforth called particles) exposed on fracture surfaces of biochars that have been incubated in soils for zero, one, and eight months. Similar results exist for all the soils investigated and for all incubation periods. For mangosteen shell biochar, as shown in Figure 8a, two factors explain 47% of the variability of the data. This low degree of explanation is indicative of the diverse nature of the mineral particles in biochar. There is a clear segregation of variables (elements) into three groups: (i) Ca, corresponding to calcite; (ii) Mn, Na, Mg, P, Cl, S, and K mostly corresponding to the more labile constituents; and (iii) Si, Al, and Ti, which correspond to soil clay that has entered particles. The factor diagram for mangosteen shell biochar, indicating the distribution of analyses (particles), shows a discrete group of analyses for

one and eight month incubated biochar, which correspond to clay. The remaining particles, including those for zero incubation time, extend along an axis that represents various mixtures of the Ca group and the Mn, Na, Mg, P, Cl, S, and K group. The analyses of the original biochar are relatively rich in the labile elements, whereas the one and eight month data are for materials with complex compositions dominated by Ca (calcite).

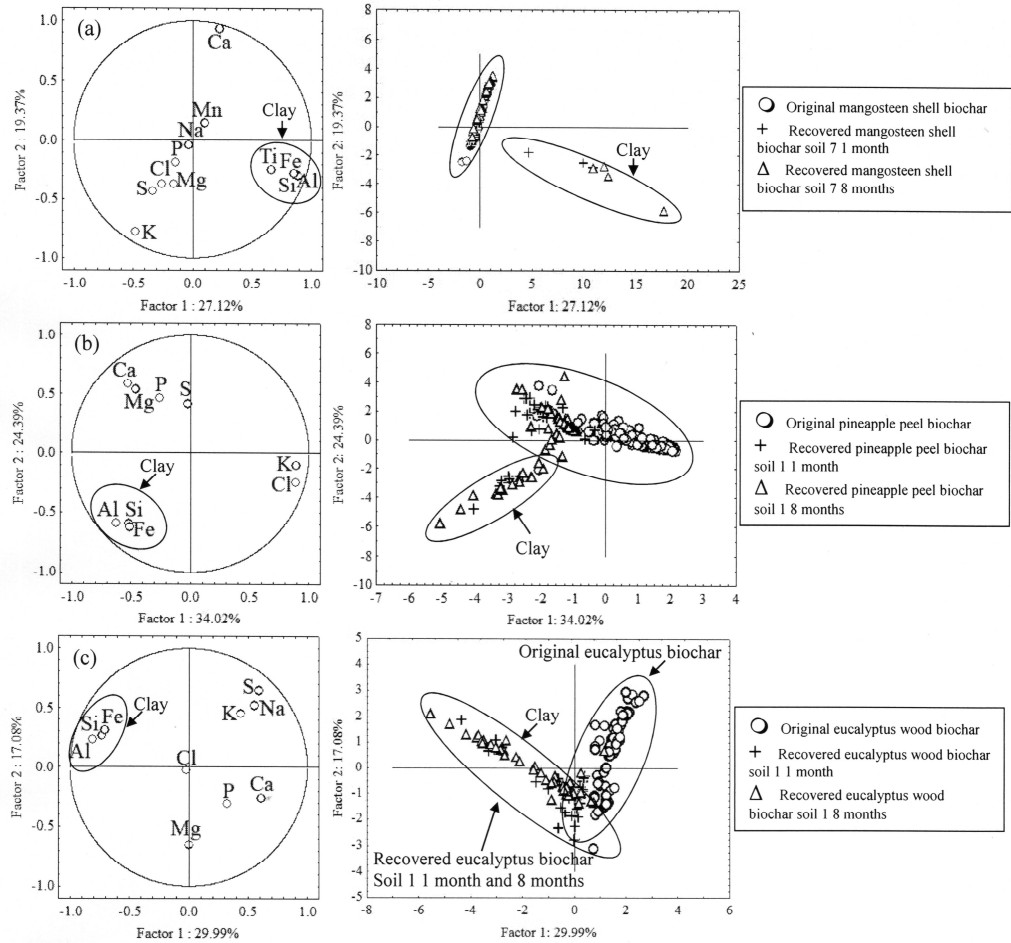

**Figure 8.** Factor plots for SEM-EDS chemical analyses of individual mineral particles in mangosteen (**a**), pineapple peel (**b**), and eucalyptus wood (**c**) biochars. The biochars had been incubated in soils for zero, one, and eight months.

The factor diagrams for pineapple peel biochar, as shown in Figure 8b, provide a simple summary of trends in chemical composition explaining 58% of the variation in data. There are three distinct affinity groups of elements which are similar but not identical to those for mangosteen shell biochar: (i) Ca, Mg, P, S (containing calcite, apatite); (ii) K, Cl (sylvite); and (iii) Si, Al, Fe (clay). Many particles in original pineapple peel biochar are rich in KCl with lesser amounts of Ca, Mg, P, and S. After incubation in the soil for one and eight months, little K remained, and the residual particles were mostly mixtures of elements dominated by the Ca, Mg, P, and S groups or a separate group dominated by introduced clay.

In the factor diagram for elements in eucalyptus biochar, as shown in Figure 8c, the clay elements (Al, Fe, Si) are a tightly associated group, as are the labile elements (K, Na, S but not Cl), whereas Ca, Mg, and P form a diffuse group. Only 47% of the variation in data is explained by the two factors. The analyses of particles from original eucalyptus biochar represent various mixtures of labile elements and the Ca, Mg, and P group. Recovered biochar particles contain many particles with diverse compositions, which correspond to mixtures of clay and the Ca, Mg, and P group. The factor analyses

for all three biochars clearly indicate that major changes in the chemical composition of particles had been induced by incubation in soil. The mobile elements (K, Na, Cl, some S) rapidly left the biochar as sylvite and other soluble salts dissolved, and the dissolved ions diffused out of biochar particles into soil. Relatively immobile elements (Ca, P, Mn, some S) were mostly retained in particles in composite particles which have complex compositions. Much soil clay and some silt-size quartz had entered the biochar particles within one month, so that the Si, Al, and Fe group becomes an important constituent of recovered biochar particles.

## 4. Discussion

The major objective of this research was to identify if biochar can replace or compliment chemical fertilizers as a source of nutrient elements for crops. At a typical application rate of 10 ton biochar per hectare, the three biochars investigated in this research would provide 55–325 kg K/ha, 48–143 kg Ca/ha, and 35–103 kg P/ha. These amounts are commonly used in, and represent, luxury rates of fertilizer application [25], and if repeated annually, might obviate the need for chemical fertilizers. This is also likely to be the situation for many of the other plant nutrient elements present in biochar. The correspondence between nutrient elements in equivalent biochar and fertilizer rates rests on the assumption that elements in biochar will be released to soil solution within the growth period of crops [26]. The current research has investigated this hypothesis and the mechanisms controlling the release of nutrient elements from biochar.

Unlike most chemical fertilizers, only a minor to moderate proportion of the K, Ca, P, and other nutrient elements in biochar are readily soluble in a simple water extraction. This is partly a consequence of nutrients being in crystals located within micropores in the insoluble, carbon matrix of biochar, which protects against dissolution within the timespan of conventional water extraction [27].

Biochar particles in moist soils have pores filled with soil solution imbibed by capillarity from the soil [28]. This solution, which may be initially acid, supports dissolution of crystals over time via diffusion of ions through pore solution to and from crystals. Thus, the release of plant nutrient elements to soil solution from within biochar particles reflects the interplay of diffusion of ions to and from crystals within fluid filled pores in biochar and the kinetics of dissolution of crystal surfaces exposed to this fluid. The compounds present in biochar may differ from those in commercial fertilizers. The net effects of these processes on the release of plant nutrient elements to soil solution has been revealed in this research and are diverse, as shown in Figures 2–4. Most K diffused from biochar particles into the soil within one month, so biochar resembles KCl and $K_2SO_4$ fertilizers in this respect [29]. Manganese showed similar behavior, with little Mn remaining in biochar after one month. Calcium and Mg exhibited quite different behaviors, with little or no loss over 8 months, and pineapple biochar extracted considerable Ca from the soil during this time. For these elements, biochar is not likely to be an effective fertilizer. Mangosteen shell biochar released about half of its P content over eight months, whereas there was no systematic loss or gain of P from pineapple peel and eucalyptus wood biochar. These different behaviors are possibly a consequence of much P in mangosteen shell biochar being in soluble potassium, magnesium, or sodium phosphates, as shown in Figure 8a, whereas P in pineapple peel and eucalyptus wood biochar is mostly in less soluble calcium phosphates, as shown in Figure 6d, Figure 7a, and Figure 8b,d.

Thus, we might regard P in pineapple peel and eucalyptus wood biochars as having an affinity with rock phosphate fertilizers, where P is present in calcium phosphate minerals (mostly apatite) that require prolonged contact with acidic soil solution to dissolve and become available to plants [11,30–32]. In contrast, the behavior of P in mangosteen shell biochar more closely resembles partially acidulated rock phosphate [30,32], where only some P is readily soluble. In general, rock phosphates and partially acidulated rock phosphates are less effective fertilizers than water soluble fertilizers such as superphosphate [30,32].

The loss of S from biochar particles is similar to that of P for mangosteen and pineapple peel biochars, whereas eucalyptus wood biochar initially contained little S but gained substantial S from

the soil during incubation, as shown in Figure 4. In contrast, commercial S fertilizers, such as gypsum, generally dissolve freely in soils [33] as they are not protected by a biochar matrix.

We can conclude from this research that K in biochar can be considered as a direct replacement for water soluble K fertilizers. This is not the case for Ca, Mg, P, and S provided by biochar, which are retained or only partly released to soil solution. Their agronomic effectiveness relative to water-soluble commercial fertilizers should be determined in glasshouse and field experiments [25]. The present work indicates that substitution of nutrients in biochar for nutrients in chemical fertilizers cannot be done without consideration of the compounds containing nutrient elements and reactions with the soil. Simple singular substitution values will not exist. Consequently, the financial benefit accrued to growers from nutrients in biochar cannot presently be established with the probable exception of potassium. An unexpected result of this research is that the extent and rates of sorption and release of nutrient elements from biochar are not sensitive to soil properties.

**Author Contributions:** Authors contributed to this article as follows: Conceptualization, Methodology and Visualization, R.J.G.; Software and Data Curation, N.P.; Formal Analysis, T.L., N.P. and R.J.G.; Investigation, T.L.; Resources, R.J.G., A.S. and I.K.; Validation, Writing-Original Draft Preparation and Writing-Review & Editing, R.J.G. and N.P.; Supervision, A.S. and I.K.; Project Administration, A.S.; Funding Acquisition, I.K.

**Funding:** The authors are very grateful for funding from the Royal Golden Jubilee Ph.D. Program of the Thailand Research Fund (6SKU52E1).

**Acknowledgments:** We would like to acknowledge the staff of the Centre for Microscopy, Characterisation and Analysis (CMCA) the University of Western Australia for SEM technical support.

**Conflicts of Interest:** The authors declare that they have no conflicts of interest.

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
