# Peer review of "Most Plant Nutrient Elements Are Retained by Biochar in Soil"

_soilsystems, doi:10.3390/soilsystems3040075_

Round 1

Reviewer 1 Report

The manuscript describes the location of nutrient elements in biochars and observed their release from biochar grains in soil, inThailand.  At a typical application rate the three biochars investigated  would provide  luxury rates of fertilizer application  that might replace the need for chemical fertilizers.

The manuscript presents important information and is well written.

Some details should be corrected (see below). Authors must check and reformulate the abstract (final sentence) and  text.

Table 1 and 2:  to explain  soil series and Soil classification system used.

Page 3 

line 73 2.3. Incubation experiment

It is a novel methodology or there is any reference, please include in this section.

line 82: 2.4. Chemical analysis. Where was performed the analysis? Please, include here.

Page 4  line111: to explain CRT

Author Response

1.      Table 1 and 2:  to explain soil series and Soil classification system used.

# We added the references in Lines 53 to 54 and Table 1.

2.      Page 3 line 73 2.3. Incubation experiment It is a novel methodology or there is any reference, please include in this section.

# We added the reference in Lines 78 to 79.

3.      Line 82: 2.4. Chemical analysis. Where was performed the analysis? Please, include here.

# We addressed it in Lines 77 to 78.

4.      Page 4 line111: to explain CRT

# We changed CRT to screen.

Reviewer 2 Report

This work reports the results Foremost, I believe that the introduction (short, without hypotheses and with only 5 references) is not sufficient for a scientific article. 

I don't believe that the conditions of preparation of the biochar (in a large cylindrical container) can guarantee the temperature of preparation. It is not clear to me how this temperature was recorded. This would hinder other groups to reproduce this work. 

Data in Tables 1 to 4 show no standard deviation.

No P values or results of an ANOVA are provided. No statistical analysis description in material and methods. I cannot recommend to proceed with a manuscript were analyses have not been replicated.

Due to the serious shortcomings (lack of replication, no hypotheses, etc.) I cannot recommend this article to be revised.

Author Response

1.    This work reports the results Foremost, I believe that the introduction (short, without hypotheses and with only 5 references) is not sufficient for a scientific article.
# The introduction is substantial and concisely identifies the objective and hypothesis of the research.

2.    I don't believe that the conditions of preparation of the biochar (in a large cylindrical container) can guarantee the temperature of preparation. It is not clear to me how this temperature was recorded. This would hinder other groups to reproduce this work.
# The method of preparation of biochar is robust and was thoroughly evaluated as indicated in our earlier papers [2], [4]. We added the reference in Lines 64 to 65.

3.    Data in Tables 1 to 4 show no standard deviation. No P values or results of an ANOVA are provided. No statistical analysis description in material and methods. I cannot recommend to proceed with a manuscript were analyses have not been replicated. Due to the serious shortcomings (lack of replication, no hypotheses, etc.) I cannot recommend this article to be revised.
# There is no need for statistical analysis of data in Tables 1 – 4 as these data represent the evaluation of experimental materials.  The experimental results have been subjected to statistical analysis (Figures 2 – 5).

Reviewer 3 Report

The manuscript titled “Most plant nutrient elements are retained by biochar in soil” investigated three biochars made of mangosteen shell, pinaple peel, and eucalyptus wood, and their characteristics to retain nutrients. In general, the manuscript is well written, and the findings are well explored and presented that provide the manuscript a good merit.
My small concern about the manuscript, that the authors did not distinguish well between their present and prior paper (published in Geoderma), other than mentioning the different timeframes. I would suggest to do some statistical analyses based on differences between biochar types and/or soil types. The experiment did not mention the number of replicates prepared by the authors (per samples not per treatments), however with so many soil types it might not be feasible for the authors to do so and opted to do one sample per monthly treatment.
I believe that after some minor adjustments, the article will be of interest of the Soil Systems readership and should be considered to be published.
Some specific/minor comments:
Please check for typos, e.g. Figure 5 title, line 272.
Introduction
At the end of the Introduction section, the authors should state their hypothesis.
Lines 118-119. Please check sentence.
Line 124. Please define which number correspond to which biochar.
Line 129. …3.3%, respectively. Please amend.
Line 133. The word “stressed” doesn’t seem to be a good choice. It might be changed to: highlighted, noted…
Lines 203-207. Sentences should be moved to Materials and methods section.
Line 222. The word “workers”, also not the best choice, I would suggests researchers or scientists.
Figures 6-7. It would be better to change the order of the lettering. I would suggest to start with month 1 and finish with month 8. Captures are confusing, could be rewritten/reworded. Figure 7 also has twice month 8 please check.
Lines 325-327. This sentence might need some rewording. The 7 month is correct?
Line 362. How often farmers might apply biochar annually?
Line 379. The word “several” is unnecessary.

Author Response

1.    Please check for typos, e.g. Figure 5 title, line 272.
# We checked them.

2.    Introduction At the end of the Introduction section, the authors should state their hypothesis.
# We added it in Lines 47 to 49.

3.    Lines 118-119. Please check sentence.
# We checked them.

4.    Line 124. Please define which number correspond to which biochar.
# We added them in Lines 133 to 136.

5.    Line 129. …3.3%, respectively. Please amend.
# We amended it in Line 141.

6.    Line 133. The word “stressed” doesn’t seem to be a good choice. It might be changed to: highlighted, noted…
# We replaced stressed by noted in Line 145.

7.    Lines 203-207. Sentences should be moved to Materials and methods section.
# We moved them to Lines 120 to 124.

8.    Line 222. The word “workers”, also not the best choice, I would suggests researchers or scientists.
# We changed workers to researchers.

9.    Figures 6-7. It would be better to change the order of the lettering. I would suggest to start with month 1 and finish with month 8. Captures are confusing, could be rewritten/reworded. Figure 7 also has twice month 8 please check.
# The choice of biochars used in Figures 6-7 and the order that data are presented in these figures reflect the materials that are being depicted and discussed in the text.  Thus the focus is on illustrating materials and not a time series such as a month 1 through to month 8 as suggested by the reviewer.  The use of two month 8 examples in Figure 7 is due to this system of presenting data (not materials vs. time series).

10.    Lines 325-327. This sentence might need some rewording. The 7 month is correct?
#We fixed it.

11.    Line 362. How often farmers might apply biochar annually?
# Farmers apply biochar at least once a year.

12.    Line 379. The word “several” is unnecessary.
# We fixed it.

Round 2

Reviewer 2 Report

I have the same concerns as before. The authors have not bothered to reply to these and this type of attitude is of utmost disrespect.

- The introduction section is uncommonly brief and does not provide a good overview of why this work is necessary. I review over 100 articles per year and I haven't found an introduction which is so brief. Saying that the intorduction is "substantial" seems a joke. Which work has been done before with biochar in tropical soils?

- The method of preparation of biochar will lead to heterogeneity in its properties and results being non-reproducible. Authors should have repeated the synthesis of biochar several times and provide replicates and a standard deviation for the properties of biochar. Not doing this is sloppy.

- Great that authors believe that their results do not need statistical tests. However, the scientific community would tend to disagree. Where is the statistical test for figures 2-5 provided? I cannot see anything about a RMANOVA in the description of materials and methods. Where are the P and F values? It is not enough to say they were done. They should be shown.

AFTER these very serious concerns (which would guarantee a rejection in most respectable journals) are addressed, I would be able to conduct a more detailed review of the article.

Please note that the journal requires reviewers to address these questions:

Are the methods adequately described?
Are the results clearly presented?
Are the conclusions supported by the results?

Statistical methods are not described.
Results are presented without statistical analyses

I don't know if conclusions supports the results, as I am unsure about the significance of the variations.

Author Response

No more revisions.